# Evaluation of Metabolic and Cardiovascular Risk Measured by Laboratory Biomarkers and Cardiopulmonary Exercise Test in Children and Adolescents Recovered from Brain Tumors: The CARMEP Study

**DOI:** 10.3390/cancers16020324

**Published:** 2024-01-11

**Authors:** Alberto Romano, Fabrizio Sollazzo, Serena Rivetti, Lorenzo Morra, Tiziana Servidei, Donatella Lucchetti, Giorgio Attinà, Palma Maurizi, Stefano Mastrangelo, Isabella Carlotta Zovatto, Riccardo Monti, Massimiliano Bianco, Vincenzo Palmieri, Antonio Ruggiero

**Affiliations:** 1Pediatric Oncology Unit, Fondazione Policlinico Universitario Agostino Gemelli IRCCS, 00168 Rome, Italy; serena.rivetti@gmail.com (S.R.); tiziana.servidei@guest.policlinicogemelli.it (T.S.); giorgio.attina@policlinicogemelli.it (G.A.); palma.maurizi@policlinicogemelli.it (P.M.); stefano.mastrangelo@policlinicogemelli.it (S.M.); antonio.ruggiero@unicatt.it (A.R.); 2Sports Medicine Unit, Fondazione Policlinico Universitario Agostino Gemelli IRCCS, 00168 Rome, Italy; fabrizio.sollazzo@guest.policlinicogemelli.it (F.S.); lorenzomorra71993@gmail.com (L.M.); isabella.zovatto@gmail.com (I.C.Z.); riccardo.monti1@unicatt.it (R.M.); massimiliano.bianco@policlinicogemelli.it (M.B.); vincenzo.palmieri@unicatt.it (V.P.); 3Dipartimento di Medicina e Chirurgia Traslazionale, Università Cattolica del Sacro Cuore, Fondazione Policlinico Universitario Agostino Gemelli IRCCS, 00168 Rome, Italy; donatella.lucchetti@unicatt.it; 4Department of Woman and Child Health and Public Health, Università Cattolica del Sacro Cuore, 00168 Rome, Italy

**Keywords:** childhood cancer survivor, cardiovascular risk, metabolic syndrome, cranial radiotherapy, brain tumor

## Abstract

**Simple Summary:**

Survivors of childhood brain tumors, because of the treatments they have undergone, have a greater risk of dying earlier from cardiovascular causes compared to the general population. The objective of this study was to evaluate differences between cardiovascular risk biomarkers and cardiopulmonary exercise test (CPET) results from childhood brain tumor survivors and healthy controls. We found worse CPET performance in survivors associated with higher endothelin-1 values compared to controls. Correlation analysis showed an inverse relationship between CPET results and leptin, emphasizing the role of radiotherapy as a possible cause in the genesis of a greater cardiovascular risk in survivors of childhood brain tumors.

**Abstract:**

In recent decades, the improvement of treatments and the adoption of therapeutic protocols of international cooperation has led to an improvement in the survival of children affected by brain tumors. However, in parallel with the increase in survival, long-term side effects related to treatments have been observed over time, including the activation of chronic inflammatory processes and metabolic alterations, which can facilitate the onset of metabolic syndrome and increased cardiovascular risk. The aim of this study was to find possible statistically significant differences in the serum concentrations of early biomarkers of metabolic syndrome and in the results of cardiopulmonary exercise testing between survivors of childhood brain tumors and healthy controls. This is a prospective and observational study conducted on a group of 14 male patients who survived childhood brain tumors compared with the same number of healthy controls. The concentrations of early metabolic syndrome biomarkers [adiponectin, leptin, TNF-α, IL-1, IL-6, IL-10, endothelin-1, apolipoprotein B, and lipoprotein (a)] were measured and a cardiopulmonary exercise test (CPET) was performed. Results: Childhood brain tumor survivors performed worse on average than controls on the CPET. Furthermore, they showed higher endothelin-1 values than controls (*p* = 0.025). The CPET results showed an inverse correlation with leptin. The differences found highlight the greater cardiovascular risk of brain tumor survivors, and radiotherapy could be implicated in the genesis of this greater cardiovascular risk.

## 1. Introduction

During the last few decades, the pediatric brain cancer survival rate increased due to the improvement in therapeutic regimens, which involve the combination of chemotherapy, radiotherapy, and supportive therapies (e.g., high dosage steroids, antibiotics, and growth factors) [1,2]. These treatments, although they facilitate recovery from cancer, have long-term toxic effects on organs and systems like the kidneys, heart, hearing, and endocrine system [3,4,5,6,7,8]. Furthermore, they also cause the activation of chronic inflammatory processes and metabolic alterations, which can facilitate the onset of metabolic syndrome and increase cardiovascular risk [9]. 

Cranial radiotherapy, the core of treatment in malignant brain cancers, can damage brain structures adjacent to the tumor. It causes hypothalamus–hypophysis axis dysfunction, which is associated with an increase in central fat accumulation and android/ginoide fat ratio with consequent release of inflammatory molecules [10]. Furthermore, cranial radiation causes growth hormone (GH) deficiency, sometimes following exposure to even low doses of radiation (e.g., 18 Gy), which is associated with elevated fasting insulin, dyslipidemia, and abdominal obesity [11]. In these patients, GH replacement worsens insulin resistance, provokes high circadian GH levels, and facilitates the production of free fatty acids [12].

Steroid therapy, the most-used supportive therapy in brain cancers, participates in the pathogenesis of metabolic syndrome. It increases the risk of type 2 diabetes and glucose intolerance, causing central fat accumulation, cytokine release, and chronic inflammation which cause DNA damage and organ failure through reactive oxygen species (ROS) and reactive nitrogen species (RNS). Therefore, liver steatosis, premature atherosclerosis, thrombosis and myocardial infarction, osteoporosis and osteopenia, neurocognitive alteration, and neuronal tissue damage occur [13].

Also, chemotherapy contributes to metabolic syndrome through various mechanisms that are difficult to distinguish because antiblastic drugs are frequently administered together. Furthermore, some drugs have known toxic effects. For example, anthracyclines cause left ventricular pathological remodeling, fibrosis, and afterload abnormalities leading to cardiovascular damage and hypertension by inducing p53, a transcription factor that regulates the cell cycle and acts as a tumor suppressor [14]. Platinum compounds cause endothelial damage, with consequent cardiovascular risk, through a cytokine-mediated mechanism, which leads to production of ROS [15]. The exposure of endothelial cells to platinum leads to endothelial release of GM-CSF, IL-6, IL-1, and IL-8. Furthermore, IL-1 can induce superoxide dismutase (SOD) from the mitochondria, which converts O_2_ to H_2_O_2_, resulting in endothelial damage [16]. All chemotherapy drugs can facilitate the onset of metabolic syndrome in common and nonspecific ways. They activate inflammatory processes through the accumulation of senescent cells and the increase of ROS [17]. Furthermore, chemotherapy drugs cause dysbiosis because of mucositis and the need for frequent use of broad-spectrum antibiotics for febrile neutropenia [18]. Alterations in the microbiome contribute to increased cytokine production and activation of inflammation, both of which facilitate the onset of metabolic syndrome [19,20].

Metabolic syndrome is a cluster of conditions that increase the risk for cardiovascular disease, and it is characterized by the presence of insulin resistance, obesity, hypertriglyceridemia, low high-density lipoprotein cholesterol, and hypertension [21].

Often, the presence of a diagnosis of metabolic syndrome identifies an already irreversible condition, and nowadays, the challenge is to identify markers that indicate the risk of developing metabolic syndrome before it occurs.

For this reason, recent studies have investigated the pathogenesis of metabolic syndrome in order to identify markers predictive of its onset, such as the adipokines adiponectin and leptin; the lipid markers lipoprotein(a) [Lp(a)] and apolipoprotein B (apoB); the cytokines tumor necrosis factor-alpha (TNF-α), interleukin 1 (IL-1), and interleukin 6 (IL-6); and the peptide endothelin (ET-1) [22].

Also, in recent years, many efforts have been made to identify instrumental tests useful for the early diagnosis of myocardial dysfunction in childhood cancer survivors (CCS), especially for early interception of the cardiological alterations caused by anthracyclines [4]. These methods include echocardiography, cardiovascular magnetic resonance, and cardiopulmonary exercise testing (CPET). Cardiovascular magnetic resonance and echocardiography are useful for evaluating cardiac function in basal conditions and can be used in the assessment of anthracycline-induced cardiac dysfunction [4]. The CPET is a non-invasive and dynamic test that, through a multiparametric analysis, allows for the integrated evaluation of cardiovascular, respiratory, and metabolic responses to physical exercise. It evaluates the functional reserve of all the systems involved in physical activity such as the heart, lungs, vascular system, and skeletal muscles. Therefore, it not only allows for the evaluation of cardiac function, mainly damaged by anthracycline therapy, but also for analysis of the relationships between the various systems that can be damaged during metabolic syndrome [23].

To date, not much information is available on cardiovascular risk in childhood brain cancer survivors, and even less is available on laboratory and instrumental methods useful for predicting the appearance of the typical elements of metabolic syndrome. The objectives of this pilot study are to analyze whether there are differences between CPET performance and concentrations of early biomarkers of metabolic syndrome and cardiovascular risk between childhood brain tumor survivors and healthy control subjects.

## 2. Materials and Methods

### 2.1. Study Endpoints

The primary endpoints were:-To analyze differences in the concentration of early biomarkers of metabolic syndrome between childhood brain tumor cancer survivors and healthy controls [adiponectin, leptin, TNF-α, IL-1, IL-6, IL-10, endothelin, apoB, and Lp(a)].-To analyze differences in CPET results between childhood brain tumor cancer survivors and healthy controls.

The secondary endpoints were:-To evaluate the correlation between early biomarkers of metabolic syndrome and the results of CPET.-To evaluate the correlation between the results of CPET and the treatments used (chemotherapy, radiotherapy, and steroid therapy).-To evaluate the correlation between early biomarkers of metabolic syndrome and the treatments used (chemotherapy, radiotherapy, and steroid therapy).

### 2.2. Study Design, Patients’ Characteristics, Ethical Approval, and Inclusion and Exclusion Criteria

This is an exploratory, pilot, prospective, and observational study involving 14 male childhood brain cancer survivors compared to 14 healthy controls matched for sex and age in the period from 20 June 2023 to 30 September 2023.

Cases were male brain cancer survivors in regular follow-up at the Pediatric Oncology Unit of Fondazione Policlinico Universitario “Agostino Gemelli” IRCCS. Inclusion criteria for the cases were: diagnosis of brain cancer during pediatric age (0–16 years); age equal to or greater than 12 years; having undergone radiotherapy (associated or not with chemotherapy); disease remission for at least 5 years (with a magnetic resonance negative for disease in the last year); absence of balance disorders and of strength deficits in the lower/upper limbs; absence of congenital or acquired cardiac disease; and Karnofsky performance status 100.

The male sex was chosen to eliminate the differences that sex would have had on CPET performance.

Healthy controls were volunteers who accessed the outpatient services of Sport Medicine of Fondazione Policlinico Universitario “Agostino Gemelli” IRCCS. Inclusion criteria were: age equal to or greater than 12 years; good state of health; no history of cancer; no more than 7 consecutive days of steroid therapy in their life; and no sports played at a professional competitive level.

Informed consent was obtained from the parents or the legal guardians of the enrolled patients or from the patients themselves for adults. The study was carried out following the Declaration of Helsinki and was approved by the ethics committee of Fondazione Policlinico Universitario “Agostino Gemelli” IRCCS (protocol ID 5729, approval letter number 0019314/23 dated 20 June 2023).

### 2.3. Disease and Treatment Related Data

For each case, the following data were collected:-Age, weight, height, BMI, and relative percentile at diagnosis;-data related to the neoplastic disease (date of diagnosis, histology, presence of metastasis at diagnosis, and primarily affected site);-type and duration of the treatment [radiotherapy (site and dose), chemotherapy, or high-dose chemotherapy with autologous transplantation (ASCT)];-data on steroid supportive therapy during treatment;-data relating to any endocrinological defects developed during oncological treatment or subsequently.

### 2.4. Personal and Family History, Anthropometric Characteristics, and Information Relating to the Level of Physical Activity

Both cases and controls underwent a medical history questionnaire including general medical history, familial history with attention to metabolic and cardiovascular disease, and physical activity level (according to the International Physical Activity Questionnaire) [24].

The International Physical Activity Questionnaire (IPAQ) comprises a set of 4 questionnaires. Long and short versions for use by either telephone or self-administered methods are available. We used the short form, which consists of 7 items to estimate the time spent performing physical activities and inactivity (available on the Internet at https://sites.google.com/view/ipaq, accessed on 29 November 2023). Based on the answers to the IPAQ questionnaire, patients and controls were divided into three degrees of activity: low level of physical activity (individuals who do not belong to the moderate or high categories and have a low level of physical activity), moderate level of physical activity (individuals who meet one of the following three criteria: 3 or more days of activity lasting more than 20 min per day OR 5 or more days of moderate physical activity or walking of at least 30 min per day OR 5 or more days of any physical activity achieving a minimum of at least 600 MET-min/week), or high level of physical activity (individuals who meet one of the following three criteria: vigorous physical activity for at least 3 days/week and accumulating at least 1500 MET-min/week OR 7 or more days of moderate or vigorous physical activity achieving a minimum of 3000 MET-min/week).

Also, all of the participants underwent a physical examination, and the following data were collected:-Height, height-for-age percentile, and Z-score. Percentiles were calculated with a percentile calculator available on Internet, based on CDC growth charts (https://peditools.org/, accessed on 13 November 2023).-Weight, weight-for-age percentile, and Z-score. Weight was measured in the absence of clothing (except undergarments) with electronic scales. Percentiles were calculated with a percentile calculator available on the Internet, based on CDC growth tables (https://peditools.org/, accessed on 13 November 2023).-BMI, BMI-for-age percentile, and Z-score. BMI was calculated using the formula (weight in kg)/(height in m)^2^. The BMI percentile and Z-score were calculated by percentile calculator available on the Internet, based on CDC growth tables (https://peditools.org/, accessed on 13 November 2023).-Waist circumference, measured at the point of smallest circumference between the last rib and the top of the iliac crest and reported in cm.-Hip circumference, measured at the major circumference point at the posterior extension of the buttocks and reported in cm.-Waist to hip ratio (WHR), measured using the formula (waist circumference in cm)/(hip circumference in cm); based on the value obtained.-Waist to height ratio (WHtR), measured by the formula (waist circumference in cm)/(height in cm); a value of WHtR > 0.5 was considered indicative of central obesity [25].− Systolic and diastolic blood pressures, reported in mmHg and pressure-per-age percentiles, calculated for patients > 18 years by comparison of CDC tables [26] and for patients <18 years by calculator available on the Internet (https://www.mdcalc.com/calc/4052/aap-pediatric-hypertension-guidelines, accessed on 13 November 2023).

### 2.5. Metabolic and Cardiovascular Risk Biomarkers

Both cases and controls underwent peripheral venous blood sampling after a 12 h fast. Measurements of glucose, triglycerides (TG), total cholesterol, high-density lipoprotein (HDL), low-density lipoprotein (LDL), were performed at the central laboratory. Also, all the cases were subjected to TSH, fT3, fT4, ACTH, GH, IGF-1, FSH, and LH testing of blood.

The following biomarkers were measured in serum samples: adiponectin, leptin, TNF-α, IL-1, IL-6, IL-10, endothelin, apoB, and Lp(a).

Serum was collected using standard operating procedures and was aliquoted (10 vials/patient) and stored at −20 °C until analysis.

Human Magnetic Luminex Screening Assays (Bio-Techne s.r.l., Milano, Italy) or ELISA assays were used according to the analytes. Specifically, three Luminex panels were used as follows. Panel 1: IL-6, IL-10, Leptin, TNF-alpha, and IL-1 beta; panel 2: adiponectin; panel 3: endothelin-1.

SimpleStep ELISA^®^ Kits (Abcam, Cambridge, UK) were used for quantitative measurement of ApoB (ab190806) and Lp(a) (ab212165) per the manufacturer’s instructions. Serum samples were diluted to 1:2000 (for ApoB) and 1:1000 (for Lp(a)).

### 2.6. Cardiopulmonary Tests

Each participant underwent CPET; before performing the CPET, all enrolled subjects underwent an echocardiogram in order to exclude the presence of unknown heart disease or myocardial function deficit.

At least 2 h of fasting and no physical exercise for at least 24 h were required.

All were tested through incremental exercise till voluntary exhaustion by an electronically braked cycle ergometer (COSMED E200 P/K Cosmed, Rome, Italy) via breath-by-breath gas analysis (COSMED Quark CPET, Cosmed, Rome, Italy). Voluntary exhaustion was defined as the inability to maintain the imposed load and pedaling frequency despite vigorous encouragement from the examiner. The test was considered valid if at least one of the criteria for achieving a VO_2_max was met within 8–12 min from the start of the test: a respiratory quotient (RQ) equal to at least 1.12, a heart rate at peak effort above 85% of maximum, or a plateau in oxygen consumption, defined as a reduced change in O_2_ (between 1.0 and 2.0 mL/kg/min).

Ramping protocols were used to individualize the test: the increase in load occurs steadily and continuously at a rate customized to the individual. The Wasserman equation will be used to determine the load increment [27]. Spirometry was performed before each CPET.

The protocol begins with 2 min of unloaded effort (3–4 watts) at a constant speed. Blood pressure and pulse oximetry are recorded every 2–3 min and at peak exercise. The peak values of the main cardiovascular, respiratory, and metabolic parameters are taken as the highest 30 s average values achieved before voluntary exhaustion of the subjects. VO_2_peak is calculated in both tests as a 15 s average of the highest VO_2_ achieved during the test. VO_2_ at aerobic threshold (AT) and ventilatory compensation point (VCP) is determined by two independent investigators using the “V-slope” method and “secondary criteria” (responses of ventilatory equivalents for CO_2_ and O_2_ (VE/VCO_2_, VE/VO_2_) and end-tidal pCO_2_ and end-tidal pO_2_ responses (PetCO_2_ and PETO_2_)).

For CPET data, percentages of predicted values for VO_2_peak were determined according to Burstein et al. in patients younger than 18 years and Wasserman equations in those older than 18 years [27,28].

### 2.7. Simple Size and Statistical Analysis

Due to the absence of data on CPET performance and early metabolic syndrome biomarkers in the study population, we set the sample size for this study at *n* = 14 patients compared to the same number of healthy controls. This number intercepts different expected proportions, with a confidence level of 95% and a margin of error ranging from a minimum of 5.21% (for an expected proportion of 1%) to a maximum of 26.19% (for an expected proportion of 50%). Also, it manages to identify differences in mean values between patients and controls in terms of effect size equal to 0.98, with a power of 80% at a significance level of 5% (calculated with the G*Power 3.1 software. 9.7.).

A descriptive analysis of all the parameters collected was performed. Quantitative variables are reported as means and standard deviations (SD). Qualitative variables are presented as frequencies and absolute percentages (%). Student’s *t* test was used for the comparison of continuous variable and the Chi squared test was used for the comparison of categorical variables. The Pearson correlation coefficient (r) was used to study the relationship between biomarkers, CPET results, and previous cancer treatment. A *p*-value < 0.05 was considered statistically significant.

All statistical analyses were performed with Jamovi V.2.2.5 software.

## 3. Results

### 3.1. Disease and Treatment Related Data

This exploratory, pilot, prospective, and observational study was conducted on 14 male childhood brain cancer survivors who underwent cranial radiotherapy compared with 14 healthy controls matched for age and sex. Data regarding the disease and treatment of patients are shown in Table 1.

At the time of diagnosis, the mean age of patients was 9.6 (SD 5). The mean weight was 43.8 kg (SD 22.3), with a mean weight percentile of 71.8 (SD 33.9); the mean height was 1.4 m (SD 0.3), with a mean height percentile of 52.8 (SD 33.3). The mean BMI was 21.1 (SD 4.8), and the mean BMI percentile was 87.9 (SD 17.3).

As to the histology of the tumors, eight patients (57%) had a germ cell tumor, four patients (29%) had a medulloblastoma, and two patients (14%) had an ependymoma. For eight patients (57%), the tumor was localized in the middle cranial fossa; for six patients (43%), it was in the posterior cranial fossa. One patient, affected by a pineal germinoma, had spinal cord metastasis at the dorsal lumbar level.

All patients were subjected to cranial radiotherapy. Twelve patients (86%) were subjected to chemotherapy. The drugs used were etoposide, ifosfamide, cisplatin, carboplatin, vincristine, and lomustine combined following different protocols.

Two patients (14%) were subjected to high-dose chemotherapy followed by an autologous stem cell transplant (ASCT), and the drugs used were thiotepa, busulfan, and cyclophosphamide. One of them had a germ-cell tumor of the pineal gland and the third ventricle, and the other had a classic medulloblastoma.

Nine (64%) of the patients had endocrinological deficiencies at the time of enrollment. Panhypopituitarism occurred in 4 (29%) patients, hypothyroidism in 2 (14%), and GH deficiency in 3 (21%) patients. The two patients with hypothyroidism had a cancer of the posterior cranial fossa, underwent whole-brain radiotherapy with boosts on focal lesions and spinal radiotherapy, and used levothyroxine for treatment. Among the patients with GH deficiency, two had a cancer of posterior cranial fossa (ependymoma and medulloblastoma) and one a germ-cell cancer of the pineal region; they all received localized radiotherapy and received GH for treatment of deficiency. Among the four patients with panhypopituitarism, two had a tumor of the pineal region and underwent localized radiotherapy (one patient takes levothyroxine and desmopressin, the other levothyroxine and hydrocortisone), one had germ-cell tumors of the diencephalon and pituitary region and underwent localized radiotherapy (levothyroxine, hydrocortisone, testosterone, and desmopressin therapy is ongoing), and one had a tumor of the posterior cranial fossa and underwent whole-brain radiotherapy with boosts on focal lesions and spinal radiotherapy. The latter took only levothyroxine for therapy (refusing GH therapy).

At the time of enrollment, the mean age of patients was 25 years. The mean of follow-up was 171 months, 14.25 years from the end of treatment.

### 3.2. Personal and Family History, Anthropometric Characteristics and Information Relating to the Level of Physical Activity of Patients and Controls

The family history of diabetes was positive for 10 (71%) patients and 9 (64%) healthy controls. The family history of cardiovascular disease was positive for 10 (71%) patients and 4 (29%) controls. Two patients (14%) and 5 (36%) controls had a family history of obesity. Two patients of 14 (14%) smoked 1–2 cigarettes/day; none of the controls smoked.

Table 2 shows the comparison of the anthropometric characteristics of patients and controls at the time of enrollment.

In the comparison of the anthropometric characteristics of patients and controls, on average, controls weighed more than cases (*p* value 0.025) and had a higher percentile by age (*p* value 0.015). Similarly, on average, controls were taller than cases (*p* value 0.015) and had a higher percentile by age (*p* value 0.007).

There were no significant differences in BMI between the two populations. Minimal differences were observed in waist and hip circumferences, although these were not statistically significant. Differences in WHR and WHtR resulted as statistically significant. In particular, WHtR resulted higher in cases than in controls, but mean WHR values resulted higher in controls than in cases.

Furthermore, five (36%) cases and three (21%) controls showed a pathological WHtR (>0.5).

Figure 1a,b shows the number of patients and controls who participate in physical activity.

Five patients (38%) versus three controls (21%) maintained usual physical activity (*p* value 0.022). The IPAQ score showed that 1 patient and 3 controls had a low level of physical activity, 8 patients and 9 controls had a moderate level of physical activity, and 5 patients and 2 controls had a high level (*p* value 0.28). 

### 3.3. Metabolic and Cardiovascular Risk Biomarkers

Table 3 shows the concentration of biomarkers of metabolic and cardiovascular risk in patients and controls.

The comparison between the lipid profile blood test of cases and controls showed an increase in the mean value of total cholesterol, triglycerides, and LDL, even if the difference were not statistically significant.

Among the experimental markers of metabolic syndrome, endothelin showed mean values higher in cases than in control (*p* value 0.025), but no subjects showed frankly pathological values of endothelin. In addition, the values of lipoprotein (a), leptin, IL 1-β, and IL-6 were higher in cases than in the control, even if these differences were not statistically significant.

Hormonal assessments were performed in cases. Three patients showed a high value of TSH, even though they were on replacement therapy with levothyroxine.

None of the subjects in the two groups met the diagnostic criteria for metabolic syndrome.

### 3.4. Cardiopulmonary Tests Results

The analysis of CPET data showed a significant difference in many items: VO_2_ peak (mL), VO_2_%, VO_2_@AT, OUES, OUES%, and VO_2_/HR showed a lower value in cases than in controls, with a *p* value < 0.05. No subjects showed frankly pathological performances on the CPET.

In Table 4 are reported the results of the cardiopulmonary tests.

### 3.5. Relation between CPET Results, Metabolic and Cardiovascular Biomarkers and Treatment

Table 5 and Table 6 show the relationship between the biomarkers of cardiovascular risk and the results of CPET.

The correlation study revealed some significant relationships. There were negative correlations between OUES and total cholesterol and between OUES and LDL; dVO_2_/WR slope and BR correlated negatively with triglycerides values. Conversely, there was a positive correlation between VE peak and glucose.

Furthermore, there were other statistically significant relationships between experimental biomarkers and CPET results. There were negative correlations between BR and Lp(a), and VO_2_ pro kg and IL-6; conversely, there was a positive correlation between VE/VCO_2_ slope and IL-1β. Particularly, leptin values correlated negatively with VO_2_ peak, VO_2_ pro kg, VO_2_%, VO_2_@AT, OUES%, VO_2_/HR, VO_2_/hr%, and peak circulatory power, such that when leptin increases, the other variable decreases, and vice-versa. On the contrary, leptin correlated positively with VE/VCO_2_ slope, such that leptin decreased as the other variable decreased, or leptin increased while the other increased. Moreover, negative correlations were noted between endothelin and OUES%, VO_2_/HR, and peak circulatory power.

Regarding CPET parameters and cancer treatment, there were positive correlations between BR and brain radiotherapy total dose and between OUES and total cyclophosphamide dose. Conversely, there was a negative correlation between VE/VCO_2_ slope and total cyclophosphamide dose.

Regarding metabolic and cardiovascular biomarkers and cancer treatment, there were positive correlations between IL-6 and length of steroid therapy, and between IL-10 and cisplatin and ifosfamide total doses. Conversely, there were negative correlations between Lp(a) and brain radiotherapy total dose, and between TNFα and cisplatin and ifosfamide total doses. Data on the correlation between biomarkers and treatment and between CPET results and treatment are showed in Appendix A.

The tables showing the relationships between the biomarkers of cardiovascular risk, the results of CPET, and the treatment utilized for each cancer type are reported in Appendix A.

## 4. Discussion

Survivors of childhood cancer have a 7-fold greater risk than the general population of dying earlier from cardiovascular causes, accounting for a quarter of all deaths within 45 years from cancer diagnosis [29]. Such a high incidence explains the need to carry out careful follow-up of patients who have survived childhood cancers in order to be able to diagnose the onset of metabolic syndrome as soon as possible and to implement measures aimed at reducing the risk of death from cardiovascular causes. In fact, to date, the strategies available for the management of metabolic syndrome and, therefore, for reducing cardiovascular risk, are represented by regular physical exercise and an adequate diet [30]. However, these strategies are effective if implemented before irreversible damage has occurred. It is therefore necessary to identify tools that allow for early diagnosis and that can also guide therapeutic strategies aimed at interrupting the mechanisms of cardiovascular damage present in this population of subjects.

For this reason, in our study, we analyzed the possible differences in biomarkers of metabolic syndrome and CPET results between childhood brain cancer survivors and healthy controls.

The brain cancer survivors enrolled in our study did not present cardiovascular and metabolic risk factors before diagnosis nor were their family histories more predisposing compared to the control population analyzed.

However, at the time of enrollment, when anthropometric data were compared between cases and controls, cases seemed to have an average weight lower than controls, but they also had an average height centile lower than controls. Therefore, the lower weight can be considered a consequence of the lower height. Regarding the mean BMI, there were no evident differences between the two groups. However, although the diagnosis of obesity is based on the BMI value, the BMI does not give indications of the distribution of body fat, which is the real metabolic risk factor. The BMI calculation is simple, but it estimates excess weight rather than excess fat [31]. Body fat distribution can be better estimated through parameters such as waist circumference and WHtR. In fact, waist circumference was greater in cases than in controls, as was WHtR, and 5 patients (36%) had a pathological WHtR value (>0.5) compared to 3 (21%) controls. An increase in pathological WHtR in CCS compared to healthy controls has already been described; in the study by Steinberger J et al., a prevalence of 24% was reported in the population of CCS compared to 11.2% in healthy controls [32]. The mean WHtR in our population (0.49) is similar to the average value of 0.48 reported in other studies and is higher than the average value of the general population (approximately 0.43). These results indicate that CCS have higher central adiposity than the non-cancer population.

Between cases and controls there was no diagnosis of metabolic syndrome, maybe because our population was too young to experience the long-term effects of risk factors in terms of items required for diagnosis. Hence the need to find early markers of metabolic syndrome.

Analysis of the lipid assessment and the biomarkers revealed a statistically significant difference only in ET-1 values between the two populations analyzed, with higher values in cases than in controls (1.3 pg/mL (SD 0.4) vs. 1 pg/mL (SD 0.2), *p* value 0.025). ET-1 is a peptide produced by the vascular endothelium that plays a vasoconstrictor and pro-inflammatory role and stimulates mitosis, cellular proliferation, free radical formation, and platelet activation [33]. Based on these actions, ET-1 is considered an important factor in the development of vascular dysfunction and cardiovascular disease depending on its role in the pathogenesis of hypertension, heart failure, and atherosclerotic vascular disease [34]. ET-1 vasoconstrictor tone is elevated in obesity and in middle-aged adults with impaired fasting blood glucose, and it seems to play a role in plasma lipid derangements and insulin resistance associated with obesity [35,36]. This contributes to diminished endothelium-dependent vasodilation and augmented cardiovascular risk [37,38]. The presence of the metabolic syndrome is associated with higher ET-1 vasoconstrictor tone in overweight and obese adults [39]. Other studies have already highlighted the possibility of an association between mortality and levels of ET-1 precursors in a population of neoplastic patients, underlining the close interdependence between malignancy and the development of cardiovascular damage [40]. Furthermore, altered values of ET-1 have been observed in endothelial disfunction subsequent to chemotherapy in acute lymphoblastic leukemia survivors, attributed mainly to radiotherapy and anthracycline-based chemotherapy [41]. Notoriously, radiotherapy also induces alterations in endothelial cell function that can manifest with endothelial cell activation, enhanced leukocyte–endothelial cell interactions, increased barrier permeability, and initiation of apoptotic pathways. [42]. However, to our knowledge, the increase in ET-1 values has not yet been observed in brain cancer survivors, and this finding confirms the high cardiovascular risk of this population.

The other biomarkers did not show a statistically significant difference between cases and controls, probably due to the small sample size. However, the cases have mean values of total cholesterol, triglycerides, and LDL evidently higher than controls. Also, Lp(a), leptin, IL 1-β, and IL-6 tend to have higher average values in cases than in controls. These differences, although not statistically significant, could herald the development of a pathological lipid assessment and of full-blown metabolic syndrome in a few years.

Lipoprotein alterations play a key role in the increased cardiovascular risk of patients affected by metabolic syndrome. Lp(a) is an LDL-like lipoprotein that contains apo(a) and apoB100 and acts as a lipid transporter. It participates in the formation of atherosclerotic plaques as it facilitates the proliferation of smooth muscle cells, the formation of foam cells, the necrotic core, and the calcification of atherosclerotic lesions and upregulates adhesion molecules [43,44]. In a group of 56,804 participants, Waldeyer et al. observed that elevated levels of Lp(a) lead to an increased risk of major coronary events and cardiovascular disease [45]. Although some studies showed an association between elevated levels of Lp(a) and the onset of metabolic syndrome, there is very limited evidence for the role of Lp(a) in the pathogenesis of metabolic syndrome in CCS [46,47]. In our study, it seems to be higher in cases than in healthy controls.

Leptin is a protein produced by the adipocytes. Its plasmatic value is influenced by gender (greater in females than in males) and is directly proportional to the amount of adipose tissue: in the case of an excess of adipose tissue, high levels of leptin are produced to inhibit the appetite and reduce food intake [48,49]. In fact, leptin acts on the solitary tract and on the ventral tegmental area in the brain stem. Here, it stimulates neurons secreting proopiomelanocortin and inhibits the orexigenic agouti-related protein/neuropeptide Y-containing (AgRP/NPY) neurons [50]. In adults, leptin is a predictor of insulin resistance, glucose intolerance, and metabolic syndrome regardless of underlying obesity [51]. Furthermore, elevated leptin levels were found to be an important predictor of cardiovascular-related death and hypertension [52]. As mentioned before, leptin acts on receptors in the arcuate nucleus and the hypothalamic nuclei to reduce appetite. Radiotherapy can damage these structures, with the onset of leptin resistance and consequent leptin overproduction by fat tissue [53]. It was demonstrated that high-dose cranial radiation (>30 Gy to the hypothalamic–pituitary axis) determines leptin resistance at hypothalamic receptors and an increase in its circulating levels [54]. The lack of leptin action on the hypothalamus causes an increase in appetite, leading to accumulation in adipose tissue, whereas the high level of leptin leads to insulin resistance and glucose intolerance [22]. So, because of leptin resistance, CCS exposed to cranial irradiation have a higher BMI, fat mass, and central obesity. It was demonstrated that, in CCS of brain tumors, plasma leptin values were higher than in healthy subjects and correlated with central fat indicators such as waist-to-height ratio and waist-to-hip ratio [55]. According to these findings, our analysis showed higher levels of leptin in cases than in controls.

In our study, IL-1β and IL-6 values also tended to be higher than in controls, and this confirms that survivors of childhood brain cancer may have a chronic inflammatory state that favors the onset of metabolic syndrome and cardiovascular risk.

In fact, IL-1β was observed to have an insulin resistance action, and its increased plasma levels, as well as IL-6, increased the risk of developing type 2 diabetes [56] and plays a role in various metabolic processes [57].

Regarding the CPET results, a substantial difference clearly emerges in terms of CPET parameters between the two groups: The survivor group showed a significant inferiority regarding all the main indices of cardiopulmonary performance (i.e., maximum absolute peak oxygen consumption (VO_2_), VO_2_ as a percentage of predicted (VO_2_%), VO_2_ at the aerobic threshold, oxygen uptake efficiency slope (OUES), OUES as a percentage of predicted (OUES%), oxygen pulse (VO_2_/HR), and peak circulatory power (PCP)) compared to healthy controls. These results occur despite the very similar levels of physical activity practiced between the two groups. In fact, the comparison of the level of physical activity measured through the IPAQ score does not denote a substantial difference between the two populations. This aspect denotes how the CCS population has already been made aware of the usefulness of physical exercise to counteract all the negative phenomena linked to physical inactivity, one of which is the development of cardiovascular risk factors. At the same time, however, this finding raises further considerations: first, that mere physical activity may not be sufficient to reverse the trend and that the training plan for these patients must be studied by a specialist and defined on the basis of CPET results in order to achieve greater effectiveness, and second, that the reduced performance is not justified solely by the state of detraining, but that other factors, cardiovascular and peripheral, intervene on it which go beyond the known myocardial dysfunction secondary to anthracycline therapy [58,59]. In fact, some studies highlighted the role played by sarcopenia, muscolar dysfunction, and reduced density of capillaries and mitochondria within myocytes in determining the altered cardiopulmonary performance in these patients [60,61]. The deepening of these aspects may provide new elements for a better understanding of cardiovascular risk in these patients.

In the literature, current evidence confirms that a high cardiovascular risk is associated with low levels of exercise tolerance, even in young people, and that a good cardiorespiratory performance represents a fundamental index of general health [62]. Some studies showed that CCS have a lower tolerance to physical exercise compared to healthy controls [63]. However, data regarding brain cancer survivors are scarce [64].

In our study, the coexistence of impaired CPET parameters with increased biomarkers values and worse anthropometric parameters in cases than in controls may suggest that CPET could be used for the early detection of subject with high risk to develop metabolic syndrome in the future. Therefore, studying exercise tolerance using CPET may represent an additional test to identify the risk of developing metabolic syndrome in CCS.

The correlation analysis shows that the CPET parameters and the biomarkers levels correlate neither with the brain RT total dose nor with steroid therapy or age, probably due to the homogeneity of the population examined. In fact, all patients received a similar dose of brain radiotherapy and of steroid therapy due to their similar pathologies and clinical features. The positive correlation between IL-10 and cisplatin and ifosfamide total doses could be explained by the direct effect of chemotherapy on triggering the release of inflammatory cytokine by the immune system [65,66].

An interesting finding in this study is the correlation between leptin and worse CPET performance. Specifically, leptin correlates negatively with maximum absolute peak oxygen consumption (VO_2_ peak), VO_2_ pro kg, VO_2_ as a percentage of predicted (VO_2_%), oxygen uptake efficiency slope as a percentage of predicted (OUES%), oxygen pulse (VO_2_/HR), VO_2_/hr%, and peak circulatory power (PCP). On the contrary, leptin correlates positively with VO_2_ at aerobic threshold (VO_2_@AT) and minute ventilation/carbon dioxide production (VE/VCO_2_) slope.

The data in the literature support these correlations because of a direct action of leptin on the heart. In fact, the literature reports the close correlation between leptin and coronary and cerebrovascular events, suggesting a direct action of leptin at the cardiac level, as leptin receptors are present on the myocardium. Furthermore, leptin modulates the release of nitric oxide (NO) and other vasodilatory agents and determines a direct activation of the sympathetic nervous system; this would explain hyperleptinemia in hypertensive subjects. [67]. Elevated levels of leptin were associated with cardiac dysfunction mediated by mechanisms such as the metabolic switch from glucose metabolism to fatty acid oxidation, which favors lipotoxicity, systemic inflammation, insulin resistance, and activation of the renin angiotensin-aldosterone, causing clinically measurable effects such as cardiac remodeling and increased fibrosis, vascular dysfunction, and inflammation [68].

Therefore, the correlation between leptin, functional parameters, and cardiovascular risk appears relevant. As described previously, radiotherapy causes leptin overproduction, and therefore, higher leptin values are found on average in CCS. In this sense, CPET could have clinical importance in the early identification of conditions compatible with an increased cardiovascular risk and different from or preceding the development of metabolic syndrome.

## 5. Conclusions

The results of this exploratory study highlight that childhood brain tumor survivors have endothelin values and CPET performance worse than the control group and that radiotherapy could contribute to increasing metabolic alterations and cardiovascular risk. This may have an important impact on the long-term prognosis because metabolic syndrome is associated with increased cardiovascular morbidity and mortality.

In our study, childhood brain cancer survivors showed worse WHtR than healthy people and tended to have worse lipid assessment and increased biomarker values compared to controls. Furthermore, they achieved worse performance on the CPET despite doing physical exercise, which is currently considered to be the best protective factor against cardiovascular risk in these patients. The relationship observed between CPET and leptin confirms the role of radiotherapy in increasing the cardiovascular risk of these patients.

Together, our data suggest that a healthy diet and consistent sport habits are not sufficient to overcome the long-term effect of radiotherapy. However, better results could probably be obtained by using CPET results to build a personalized physical activity plan.

Limitations of our study are represented by:small sample size;impossibility to establish with certainty whether the results observed are linked exclusively to the effect of long-term radiotherapy or to other factors linked to the disease or to the other treatments carried out.

Further studies are required to confirm ours and in order to delve deeper into the pathogenic mechanisms of metabolic syndrome and identify both early diagnosis tools and possible treatments.

## Figures and Tables

**Figure 1 cancers-16-00324-f001:**
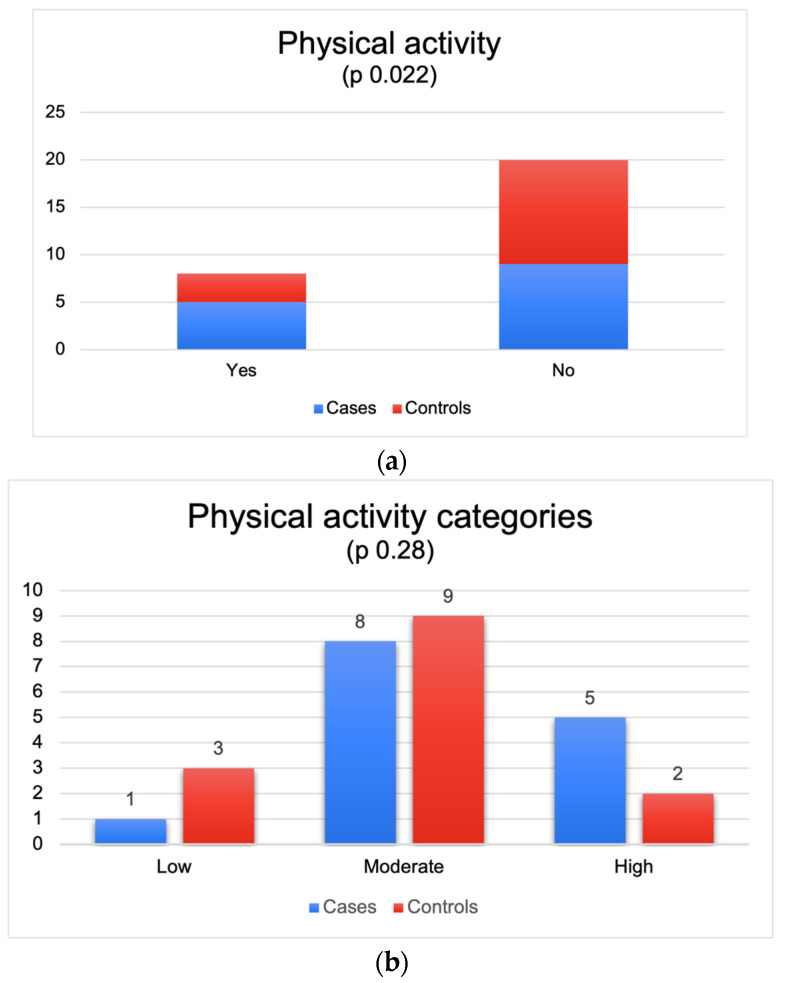
(**a**) Numbers of patients and controls who participate in physical activity. (**b**) Numbers of patients and controls who participate in physical activity.

**Table 1 cancers-16-00324-t001:** Disease- and treatment-related data (*n* = 14).

	Number of Patients (%) or Mean (SD)
Age at diagnosis (years)	9.6 (5)
Weight (kg) *	43.8 (22.3)
Weight percentile *	71.8 (33.9)
Weight Z-score *	0.9 (1.2)
Height (m) *	1.4 (0.3)
Height percentile *	52.8 (33.3)
Height Z-score *	0.1 (1.2)
BMI (kg/cm^2^) *	21.1 (4.8)
BMI percentile *	87.9 (17.3)
BMI Z-score *	0.8 (1.6)
Histology	
Germ cell tumor	8 (57%)
Medulloblastoma	4 (29%)
Ependymoma	2 (14%)
Primary localization	
Anterior cranial fossa	0 (0%)
Middle cranial fossa	8 (57%)
Posterior cranial fossa	6 (43%)
Presence of metastasis at diagnosis	1 (7%)
Patients subjected to cranial radiotherapy	14 (100%)
Patients subjected to spinal radiotherapy	5 (36%)
Cranial radiotherapy dose (Gy)	56.7 (16.6)
Spinal radiotherapy dose (Gy)	30 (0)
Patients subjected to chemotherapy	12 (86%)
Patients subjected to steroid therapy (more than 14 days)	10 (71%)
Patients subjected to ASCT	2 (14%)
Relapsed disease	2 (14%)
Endocrinological deficiencies	9 (64%)
Panhypopituitarism	4 (29%)
Hypothyroidism	2 (14%)
GH deficiency	3 (21%)
Age at the enrollement (years)	24.9 (3.9)
Time of follow up at the enrollement (months)	171 (54)

* at diagnosis. ASCT: autologous stem cell transplant; BMI: body mass index; GH: growth factor.

**Table 2 cancers-16-00324-t002:** Comparison of the anthropometric characteristics of patients and controls at the time of enrollment.

Variables	Cases [Mean (SD) or Number of Patients (%)]	Controls [Mean (SD) or Number of Patients (%)]	*p* Value
Age (years)	24.93 (3.89)	24.64 (2.92)	0.834
Weight (kg)	67.89 (10.89)	76.81 (8.85)	0.025 *
Weight percentile	48.23 (19.08)	64.79 (23.91)	0.015 *
Weight Z-score	−0.31 (1.12)	0.41 (0.70)	0.347
Height (m)	1.68 (0.10)	1.77 (0.08)	0.015 *
Height percentile	28.67 (19.65)	51.79 (33.23)	0.007 *
Height Z-score	−1.19 (1.40)	0.06 (1.13)	0.434
BMI (kg/m^2^)	23.99 (3.22)	24.59 (3.19)	0.624
BMI percentile	57.34 (26.90)	58.91 (26.29)	0.877
BMI Z-Score	0.11 (1.12)	0.31 (0.83)	0.607
Waist circumference (cm)	83.04 (8.31)	80.36 (9.83)	0.079
Hip circumference (cm)	91.07 (7.44)	87.86 (12.08)	0.178
WHR	0.91 (0.05)	0.92 (0.07)	<0.002 *
WHtR	0.49 (0.05)	0.45 (0.06)	0.049 *
WHtR > 0.5	5 (36%)	3 (21%)	0.402
BP			
Systolic (mmHg)	124.43 (14.55)	123.50 (7.76)	0.417
Diastolic (mmHg)	70.71 (7.44)	71.71 (7.61)	0.363
HR (bpm)	69.07 (11.65)	73.57 (10.25)	0.143

Student’s *t* test was used for the comparison of continuous variables and the Chi squared test was used for the comparison of categorical variables. * *p* value < 0.05. BMI: body mass index; BP: blood pressure; HR: hearth rate; SD: standard deviation; WHR: waist/hip circumference ratio; WHtR: waist/height ratio.

**Table 3 cancers-16-00324-t003:** Comparison of biomarkers of metabolic and cardiovascular risk between cases and controls.

Variables	Cases [Mean (SD)]	Controls [Mean (SD)]	*p* Value
Glucose (mg/dL)	83.36 (5.80)	75.21 (14.06)	0.056
Total cholesterol (mg/dL)	166.01 (55.78)	158.57 (29.44)	0.191
Triglycerides (mg/dL)	92.64 (50.19)	85.21 (37.90)	0.662
HDL (mg/dL)	49.21 (20.09)	48.57 (8.30)	0.913
LDL (mg/dL)	105.50 (28.05)	93.00 (25.61)	0.229
Lp(a) (mg/dL)	58.52 (71.46)	35.47 (53.10)	0.342
Apo B (mg/dL)	19.98 (8.18)	20.91 (9.13)	0.769
Leptin (ng/mL)	6.22 (4.79)	4.26 (4.84)	0.290
TNFα (pg/mL)	3.13 (1.01)	3.30 (0.84)	0.633
IL 1-β (pg/mL)	0.78 (0.46)	0.49 (0.29)	0.179
IL-10 (pg/mL)	0.36 (0.33)	0.48 (0.46)	0.568
IL-6 (pg/mL)	0.60 (0.64)	0.22 (0.33)	0.176
Endothelin-1 (pg/mL)	1.28 (0.41)	0.99 (0.17)	0.025 *
Adiponectin (μg/mL)	10.91 (3.57)	10.91 (3.47)	0.998

Student’s *t* test was used for the comparison of variables. * *p* value < 0.05. HDL: high-density lipoprotein; IL: interleukin; LDL: low-density lipoprotein; SD: standard deviation; TNF: tumor necrosis factor; Apo B: apolipoprotein B; Lp(a): lipoprotein (a).

**Table 4 cancers-16-00324-t004:** Comparison of cardiopulmonary tests results between patients and controls.

Variables	Cases [Mean (SD)]	Controls [Mean (SD)]	*p* Value
VO_2_ peak (mL)	1974.4 (451.31)	2470.9 (384.94)	0.002 *
VO_2_ pro kg	29.3 (5.25)	32.6 (5.39)	0.054
VO_2_%	61.0 (11.2)	69.7 (12.6)	0.032 *
VO_2_@AT	974.8 (238.33)	1198.8 (271.2)	0.024 *
OUES	1744.9 (684.16)	2432.1 (450.12)	0.002 *
OUES%	62.1 (15.23)	72.1 (10.7)	0.027 *
VO_2_/HR	11.0 (2.63)	13.6 (2.13)	0.005 *
VO_2_/hr%	67.2 (11.9)	74.3 (12.6)	0.069
dVO_2_/WR slope	10.6 (1.29)	10.4 (0.9)	0.611
VE/VCO_2_ slope	27.4 (2.85)	28.1 (4.85)	0.572
BR	31.9 (25.1)	41.2 (15.2)	0.124
FR peak	46.0 (10.2)	40.7 (10.2)	0.911
VE peak	85.8 (21.0)	96.2 (34.3)	0.172
PCP	331,341.4 (89,143.1)	433,262.1 (83,148.8)	0.002 *

Student’s *t* test was used for the comparison of continuous variables. * *p* value < 0.05. AT: anaerobic threshold; BR: breathing reserve; FR: frequency rate; HR: hearth rate; OUES: oxygen uptake efficiency slope; PCP: peak circulatory power; SD: standard deviation; VE: ventilation rate; VE/VCO_2_: minute ventilation/carbon dioxide production; VO_2_: pulmonary oxygen uptake; WR: work rate.

**Table 5 cancers-16-00324-t005:** Correlation between CPET results and glucose and lipid assessment.

Variables		Glucose (mg/dL)	TC (mg/dL)	TG (mg/dL)	HDL (mg/dL)	LDL (mg/dL)
VO_2_ peak (mL)	r	0.004	−0.163	0.15	−0.012	−0.24
	*p* value	0.985	0.408	0.447	0.952	0.219
VO_2_ pro kg	r	0.039	−0.06	−0.039	0.177	−0.175
	*p* value	0.845	0.842	0.842	0.369	0.374
VO_2_%	r	0.073	−0.079	0.021	0.198	−0.231
	*p* value	0.713	0.691	0.915	0.423	0.3
VO_2_@AT	r	−0.052	−0.019	0.341	−0.159	−0.019
	*p* value	0.81	0.93	0.103	0.458	0.928
OUES	r	−0.151	−0.420 *	−0.175	0.141	−0.508 *
	*p* value	0.444	0.026 *	0.372	0.474	0.006 *
OUES%	r	0.038	−0.180	0.005	0.068	−0.252
	*p* value	0.849	0.358	0.978	0.731	0.195
VO_2_/HR	r	−0.034	−0.230	0.097	−0.002	−0.303
	*p* value	0.862	0.240	0.623	0.994	0.117
VO_2_/hr%	r	0.047	−0.164	−0.018	0.131	−0.273
	*p* value	0.813	0.405	0.928	0.507	0.159
dVO_2_/WR slope	r	0.011	−0.131	−0.380 *	0.053	−0.073
	*p* value	0.955	0.506	0.046 *	0.788	0.712
VE/VCO_2_ slope	r	−0.070	0.227	0.029	−0.098	0.309
	*p* value	0.722	0.245	0.884	0.621	0.110
BR	r	−0.269	−0.076	−0.466 *	0.262	−0.012
	*p* value	0.167	0.701	0.012 *	0.178	0.953
FR peak	r	0.356	0.152	0.351	0.210	0.131
	*p* value	0.063	0.440	0.067	0.284	0.506
VE peak	r	0.495 *	0.106	0.365	0.005	−0.021
	*p* value	0.007 *	0.591	0.056	0.982	0.917
PCP	r	0.014	−0.147	0.182	−0.012	−0.224
	*p* value	0.944	0.454	0.354	0.953	0.253

The Pearson correlation coefficient (r) was used to study the relationship between variables. * *p* value < 0.05. AT: anaerobic threshold; BR: breathing reserve; FR: frequency rate; HDL: high-density lipoprotein; HR: hearth rate; LDL: low-density lipoprotein; OUES: oxygen uptake efficiency slope; PCP: peak circulatory power; SD: standard deviation; TC: total cholesterol; TG: triglycerides; VE: ventilation rate; VE/VCO_2_: minute ventilation/carbon dioxide production; VO_2_: pulmonary oxygen uptake; WR: work rate.

**Table 6 cancers-16-00324-t006:** Correlation between CPET results and metabolic and cardiovascular biomarkers.

Variables		Lp (a) (mg/dL)	Apo B (mg/dL)	Leptin (ng/mL)	TNF α (pg/mL)	IL−1β (pg/mL)	IL−10 (pg/mL)	IL−6 (pg/mL)	ET−1 (pg/mL)	AN (μg/mL)
VO_2_ peak (mL)	r	0.074	0.302	−0.479 *	0.045	−0.426	0.175	−0.320	−0.355	0.011
	*p* value	0.708	0.126	0.010 *	0.818	0.100	0.501	0.196	0.075	0.955
VO_2_ pro kg	r	0.278	0.187	−0.761 *	−0.126	−0.379	−0.011	−0.477 *	−0.164	0.205
	*p* value	0.152	0.351	0.001 *	0.522	0.147	0.968	0.045 *	0.422	0.295
VO_2_%	r	0.231	0.195	−0.727 *	−0.122	−0.414	−0.059	−0.456	−0.161	0.1
	*p* value	0.237	0.330	0.001 *	0.537	0.111	0.822	0.057	0.432	0.36
VO_2_@AT	r	−0.107	0.365−	−0.434 *	−0.180	−0.412	0.313	−0.030	−0.262	−0.091
	*p* value	0.618	0.087	0.034 *	0.401	0.184	0.256	0.915	0.228	0.471
OUES	r	0.047	0.288	−0.361	0.121	−0.363	0.143	−0.115	−0.256	0.266
	*p* value	0.811	0.145	0.059	0.540	0.167	0.583	0.649	0.207	0.171
OUES%	r	0.117	0.219	−0.575 *	−0.087	−0.357	0.028	−0.411	−0.425 *	0.074
	*p* value	0.553	0.272	0.001 *	0.659	0.174	0.915	0.090	0.014 *	0.71
VO_2_/HR	r	−0.000	0.270	−0.407 *	0.091	−0.379	0.093	−0.275	−0.776 *	−0.045
	*p* value	1000	0.174	0.031 *	0.646	0.147	0.722	0.269	0.045 *	0.809
VO_2_/hr%	r	0.124	0.176	−0.692 *	−0.088	−0.399	−0.135	−0.453	−0.262	0.06
	*p* value	0.529	0.380	0.001 *	0.657	0.126	0.605	0.059	0.197	0.726
dVO_2_/WR slope	r	−0.061	0.022	−0.199	−0.031	0.117	0.325	0.085	−0.02	0.175
	*p* value	0.757	0.913	0.310	0.876	0.665	0.203	0.738	0.994	0.37
VE/VCO_2_ slope	r	−0.200	−0.126	0.394 *	0.219	0.556 *	0.059	0.275	0.279	−0.098
	*p* value	0.308	0.532	0.038 *	0.263	0.025 *	0.822	0.269	0.168	0.62
BR	r	−0.377 *	0.019	0.016	−0.211	−0.042	−0.031	−0.139	0.265	−0.231
	*p* value	0.048 *	0.924	0.938	0.282	0.879	0.905	0.583	0.191	0.236
FR peak	r	0.316	−0.121	0.133	0.082	−0.002	0.216	0.204	−0.365	0.015
	*p* value	0.101	0.549	0.501	0.677	0.993	0.405	0.417	0.066	0.934
VE peak	r	0.187	0.246	−0.189	0.203	0.021	0.254	0.029	−0.095	0.597
	*p* value	0.342	0.215	0.336	0.300	0.938	0.325	0.910	0.644	0.288
PCP	r	0.036	0.259	−0.425 *	0.065	−0.445	0.103	−0.317	−0.397 *	0.635
	*p* value	0.854	0.192	0.024 *	0.741	0.084	0.694	0.200	0.045 *	0.25

The Pearson correlation coefficient (r) was used to study the relationship between variables. * *p* value < 0.05. AN: adiponectin; Apo: apolipoprotein; AT: anaerobic threshold; BR: breathing reserve; ET: endothelin; FR: frequency rate; HR: hearth rate; IL: interleukin; Lp: lipoprotein; OUES: oxygen uptake efficiency slope; PCP: peak circulatory power; TNF: tumor necrosis factor; VE: ventilation rate; VE/VCO_2_: minute ventilation/carbon dioxide production; VO_2_: pulmonary oxygen uptake; WR: work rate.

## Data Availability

Data are not available due to privacy restriction.

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
