# Peer review of "Evaluation of Metabolic and Cardiovascular Risk Measured by Laboratory Biomarkers and Cardiopulmonary Exercise Test in Children and Adolescents Recovered from Brain Tumors: The CARMEP Study"

_cancers, 2024, doi:10.3390/cancers16020324_

Round 1
Reviewer 1 Report
Comments and Suggestions for Authors
The authors present a topic that is important in childhood cancer survivors (CCS), namely identifying and studying variables which may predict adverse cardiometabolic late effects. In particular, use of CPET is not common in clinical practice and understanding its utility and correlative findings can add to the current methods by which CCS are evaluated. The correlation between various aspects of the CPET and biochemical markers of inflammation are interesting and important to build on.
There are several points which I would ask the authors to answer or address:
1. Abstract lines 36-37, the statement "radiotherapy is implicated in the genesis of this greater cardiovascular risk" was not identified in this paper, because there was no comparison between brain tumor survivors who were treated with radiation and those who were not. Other studies of CCS comparing those who did or did not receive CRT have shown CRT to be a risk factor for metabolic syndrome, so perhaps that is what the authors are referring to, but that should be an introduction point not a conclusion of this paper in my opinion.
2. Introduction lines 57-58 "cranial radiation causes GH deficiency, independent of radiation dose" -please clarify, as there is a clear association between RT dose and risk of HP axis dysfunction, including GH deficiency.
3. Introduction lines 149-149 "Healthy controls were volunteers who accessed to the unit of the sports medicine of Fonzadione Policlinico Universitario". What is this area where the healthy controls are recruited? What is the relevance of the sports medicine unit?
4. Among the cases, two patients were smokers and two had been treated with busulfan. Three questions here: (a) were these the same two patients or two different patients? (b) there are known effects on lung function of both cigarette smoking and busulfan therapy - would CPET identify these effects the same way that pulmonary function tests do? (c) should these patients be excluded from the study? Why or why not?
5. Most readers are likely not familiar with normative data of CPET testing, so for table 4 it may help to know whether the cases only had lower values on various parameters of the test than controls but both groups were within normal range or did the cases actually have abnormal CPET test parameters?
6. In tables 5 and 6, the data are presented for the cases only, is that correct? If blood tests were done for both cases and controls, these data should also be analyzed for the healthy controls. Was that done?
7. Various studies demonstrate various "normal" levels of endothelin 1, and are often reported relative to obesity or non-obesity and normotensive versus hypertensive. The levels range from 0.7-5 pg/mL in various reports The mean levels in both cases and controls here are within a normative range, would you agree with that statement? If so, that should be stated in the paper to add context to the findings.
8. Discussion line 572, "CPET early detects a condition which will manifest clearly later" - please clarify the meaning of this statement. This is a study of CPET in a single instance of CCS who do not have metabolic syndrome, as stated by the authors. Does this sentence intend to suggest that these CPET findings are predictive of future outcome? Please consider revising, perhaps that these findings MAY help predict future metabolic syndrome )or whatever outcome yo feel they may reflect).
9. Conclusion lines 607-610 "The results of this exploratory study highlight that childhood brain tumors survivors have an increased risk of developing metabolic syndrome and cardiovascular disease compared to their siblings and radiotherapy contributes to increasing metabolic alterations and cardiovascular risk". A few comments here (a) the authors state that the patients in this study did not have metabolic syndrome so how can the study highlight that these brain tumor survivors have an increased risk of developing an outcome they did not develop? (b) why use of the term "SIBLINGS" as a comparator group? Were the controls in the study siblings? (c) all subjects in this study received radiotherapy so the study cannot demonstrate that radiotherapy contributed to the findings, at least not in this study.
10. General comment - a control group that has not had radiation therapy but had equally intense chemotherapy (i.e. sarcoma or leukemia survivors) may serve as a better control if radiation were a factor of specific interest as potentially contributing to these findings.
Overall I believe these findings are interesting and of value to both the CCS and their providers in survivorship care programs. Reporting them in the proper context is important.
Comments on the Quality of English LanguageOverall the paper is easily understandable and English language use is good. There are instances in the discussion where various implications or conclusions are repeatedly stated, and some formatting may make the flow easier to read and less repetitive sounding.
Author Response
The authors present a topic that is important in childhood cancer survivors (CCS), namely identifying and studying variables which may predict adverse cardiometabolic late effects. In particular, use of CPET is not common in clinical practice and understanding its utility and correlative findings can add to the current methods by which CCS are evaluated. The correlation between various aspects of the CPET and biochemical markers of inflammation are interesting and important to build on.
Dear reviewer, thank you for your comments. We have revised the article following your advice. We hope that the changes we have made have improved the quality of our study.
There are several points which I would ask the authors to answer or address:
- Abstract lines 36-37, the statement "radiotherapy is implicated in the genesis of this greater cardiovascular risk" was not identified in this paper, because there was no comparison between brain tumor survivors who were treated with radiation and those who were not. Other studies of CCS comparing those who did or did not receive CRT have shown CRT to be a risk factor for metabolic syndrome, so perhaps that is what the authors are referring to, but that should be an introduction point not a conclusion of this paper in my opinion.
As observed by the reviewer, we cannot say that radiotherapy is the cause of the greater cardiovascular risk, however we can hypothesize it since radiotherapy is the factor that all patients have in common (all received radiotherapy, some also received chemotherapy and steroid therapy). For this reason we have modified the sentence as follows: “The differences found highlight the greater cardiovascular risk of brain tumor survivors and radiotherapy could be implicated in the genesis of this greater cardiovascular risk.”
- Introduction lines 57-58 "cranial radiation causes GH deficiency, independent of radiation dose" -please clarify, as there is a clear association between RT dose and risk of HP axis dysfunction, including GH deficiency.
We made a mistake in constructing the sentence. As reported by Sklar et al (citation number 11), GH deficiency can also appear following exposure to relatively low doses (e.g. 18 Gy) but the risk increases as the dose increases. By "regardless of the dose" we meant that GH deficiency can appear even at low doses; To make the concept more understandable we have modified the sentence as follows: “Furthermore, cranial radiation causes growth hormone (GH) deficiency, sometimes even following exposure to low doses of radiation (e.g., 18 Gy), which is associated with elevated fasting insulin, abdominal obesity, and dyslipidemia [11]”.
- Introduction lines 149-149 "Healthy controls were volunteers who accessed to the unit of the sports medicine of Fonzadione Policlinico Universitario". What is this area where the healthy controls are recruited? What is the relevance of the sports medicine unit?
The area was that of the sports medicine outpatiens. Sports medicine contributed to the study by performing CPETs of all subjects (both patients and controls). We changed the sentence as follows: “Healthy controls were volunteers who accessed to the outpatients of Sport Medicine of Fondazione Policlinico Universitario “Agostino Gemelli” IRCCS.”
- Among the cases, two patients were smokers and two had been treated with busulfan. Three questions here: (a) were these the same two patients or two different patients? (b) there are known effects on lung function of both cigarette smoking and busulfan therapy - would CPET identify these effects the same way that pulmonary function tests do? (c) should these patients be excluded from the study? Why or why not?
a) They were not the same patients.
b) Both busulfan and cigarette smoking can cause lung damage. In our study we did not investigate the combined effect of the two factors on the lung (the two patients who reported smoking were not the same ones exposed to busulfan). CPET investigates the three physiological systems involved in carrying out a physical exercise (pulmonary, cardiovascular and musculoskeletal system), managing to distinguish a which of these is attributable to the reduced tolerance to effort; but it does not allow analyzing the fine function of the lungs as in spirometry.
c) None of the patients in question reported respiratory symptoms (dyspnea, cough, asthma). Furthermore, the two subjects who smoke reported that they smoked a maximum quantity of 2 cigarettes per day and that this habit was recent and not long-lasting. For this reason, the subjects were not excluded from the study but they were strongly advised not to smoke.
- Most readers are likely not familiar with normative data of CPET testing, so for table 4 it may help to know whether the cases only had lower values on various parameters of the test than controls but both groups were within normal range or did the cases actually have abnormal CPET test parameters?
Cases had lower values ​​than controls. None showed frankly pathological values ​​(the study was conducted on fairly young subjects with the aim of identifying early markers of cardiovascular risk). In line 385 we added the following sentence: “No subject showed frankly pathological performances on the CPET.”
- In tables 5 and 6, the data are presented for the cases only, is that correct? If blood tests were done for both cases and controls, these data should also be analyzed for the healthy controls. Was that done?
Blood tests were executed in patients and in controls. The correlation study was also carried out for healthy subjects (for the correlations between the biomarkers and the CPET results).
- Various studies demonstrate various "normal" levels of endothelin 1, and are often reported relative to obesity or non-obesity and normotensive versus hypertensive. The levels range from 0.7-5 pg/mL in various reports The mean levels in both cases and controls here are within a normative range, would you agree with that statement? If so, that should be stated in the paper to add context to the findings.
No subject enrolled in the study presented frankly pathological values. We have specified this in the text (line 374 “but no subject showed frankly pathological values ​​of endothelin.”)
- Discussion line 572, "CPET early detects a condition which will manifest clearly later" - please clarify the meaning of this statement. This is a study of CPET in a single instance of CCS who do not have metabolic syndrome, as stated by the authors. Does this sentence intend to suggest that these CPET findings are predictive of future outcome? Please consider revising, perhaps that these findings MAY help predict future metabolic syndrome )or whatever outcome yo feel they may reflect).
We agree with the reviewer. Our study is a pilot study and the sentence we wrote is poorly constructed. We rewrote the sentence as follows: “In our study, the coexistence of impaired CPET parameters with increased biomarkers values and worse anthropometric parameters in cases than in controls may suggest that CPET could early detects subject with high risk to develop metabolic syndrome in future. Therefore, studying exercise tolerance using CPET may represent an additional test to identify the risk of developing metabolic syndrome in CCS.”
- Conclusion lines 607-610 "The results of this exploratory study highlight that childhood brain tumors survivors have an increased risk of developing metabolic syndrome and cardiovascular disease compared to their siblings and radiotherapy contributes to increasing metabolic alterations and cardiovascular risk". A few comments here (a) the authors state that the patients in this study did not have metabolic syndrome so how can the study highlight that these brain tumor survivors have an increased risk of developing an outcome they did not develop? (b) why use of the term "SIBLINGS" as a comparator group? Were the controls in the study siblings? (c) all subjects in this study received radiotherapy so the study cannot demonstrate that radiotherapy contributed to the findings, at least not in this study.
We modified the sentence making it more in line with the results observed. We also specified that radiotherapy could be implicated in the genesis of these results (we do not intend to state this with certainty, but we are only hypothesizing as all the subjects underwent radiotherapy).
Now the sentence is “The results of this exploratory study highlight that childhood brain tumors survivors have endothelin values ​​and CPET performance worse than the control group and that radiotherapy could contributes to increasing metabolic alterations and cardiovascular risk. This may have an important impact on the long-term prognosis because the metabolic syndrome is associated with increased cardiovascular morbidity and mortality.”
- General comment - a control group that has not had radiation therapy but had equally intense chemotherapy (i.e. sarcoma or leukemia survivors) may serve as a better control if radiation were a factor of specific interest as potentially contributing to these findings.
Thanks for this advice. In light of these results we hope to be able to further deepen our research on this population and we will try to carry out a comparison with subjects who did not undergo radiotherapy but only chemotherapy.
Overall I believe these findings are interesting and of value to both the CCS and their providers in survivorship care programs. Reporting them in the proper context is important.
Thanks so much for these tips. We hope we have managed to improve our article.

Reviewer 2 Report
Comments and Suggestions for Authors
Dear Authors
The study entitled 'Evaluation of metabolic and cardiovascular risk measured by 2 laboratory biomarkers and cardiopulmonary exercise test in 3 children and adolescents recovered from brain tumors: the 4 CARMEP study' was reviewed and the following remarks should be addressed.
1. Abstract: Line-21- the word thank is not suitable in scientific writing-please rethink and rephrase it.
2. Line 33- Result- "c" of childhood should be in capital.
3. Line 35- P-value should be written as P=0.025.
Manuscript:
Line- 46-49---Split, rephrase and make it simple for reader.
Line-50---Keep name of systems first then organs name.
Line-73--- What is P53---mention about it in 3-4 words.
Line-117- Study endpoints are the part of method section.
Line-286---Is there any specific reason to use JAMOVI software for statistical analysis, i mean why not- SPSS or STATA.
In all the tables and figures - All the statistical tests applied and other points should be mentioned in the legend. The table and figures should be self-explanatory.
I strongly recommend to recheck the P-value mentioned in the tables (Table 1 particularly-for AGE). P<0.002?
Please verify the P-Value in all the tables.
What is the meaning of the star mark/Asterisk?
In line--212, 462, 352---what is the meaning of >0.5.
Important: There are many limitations of the study that must be described well in the manuscript.
Conclusion - Make it more targeted and precise.
Comments on the Quality of English Language
Dear Authors
Please review the manuscript for English language/ typographical errors.
Author Response
Dear Authors
The study entitled 'Evaluation of metabolic and cardiovascular risk measured by 2 laboratory biomarkers and cardiopulmonary exercise test in 3 children and adolescents recovered from brain tumors: the 4 CARMEP study' was reviewed and the following remarks should be addressed.
- Abstract: Line-21- the word thank is not suitable in scientific writing-please rethink and rephrase it.
We rephrase it as follows: “In recent decades, the improvement of treatments and the adoption of therapeutic protocols of international cooperation has led to an improvement in the survival of children affected by brain tumors”
- Line 33- Result- "c" of childhood should be in capital.
We corrected it.
- Line 35- P-value should be written as P=0.025.
We corrected it.
Manuscript:
Line- 46-49---Split, rephrase and make it simple for reader.
We rephrase it as follows: “During the last decades, paediatric brain cancer survival rate increased due to the improvement in therapeutic regimens, which involve the combination of chemotherapy, radiotherapy and supportive therapies (e.g., high dosage steroids, antibiotics, and growth factors).”
Line-50---Keep name of systems first then organs name.
We corrected it: “These treatments, although allowing the recovery from cancer, cause long-term toxic effects on organs and systems like kidneys, heart, hearing and endocrine system [3-8].”
Line-73--- What is P53---mention about it in 3-4 words.
We modified in: “anthracyclines cause left ventricular pathological remodelling, fibrosis, and afterload abnormalities leading to cardiovascular damage and hypertension, by inducing p53, a transcription factor that regulates the cell cycle and acts as a tumor suppressor [14].”
Line-117- Study endpoints are the part of method section.
In the original version of the article, the study endpoints were within the materials and methods. They have been moved to the end of the introduction at the editor's request.
Line-286---Is there any specific reason to use JAMOVI software for statistical analysis, i mean why not- SPSS or STATA.
No, we used JAMOVI because we are more confident in its use.
In all the tables and figures - All the statistical tests applied and other points should be mentioned in the legend. The table and figures should be self-explanatory.
In paragraph 2.6. all statistical tests used in data analysis are reported. As required by the rules of the journal, the explanations of all the acronyms are given in the legend of the tables.
I strongly recommend to recheck the P-value mentioned in the tables (Table 1 particularly-for AGE). P<0.002?
Please verify the P-Value in all the tables.
Table 1 shows the descriptive data of the patients (no comparisons are reported and therefore there are no p values ​​to report). Table 2 shows an error in the p value of the age comparison that we corrected. We checked all the p values ​​reported in the article.
What is the meaning of the star mark/Asterisk?
We marked with a * all the statistically significant results (p value <0.05). We specified it into the table legends.
In line--212, 462, 352---what is the meaning of >0.5.
WHtR is the waist circumference in cm)/(height in cm. When it is >0.5, it is indicative of central adiposity (as reported in line 213) and is considered pathological as indicated in reference 25.
Important: There are many limitations of the study that must be described well in the manuscript.
Conclusion - Make it more targeted and precise.
In accordance with the indications of reviewer 1, we modified the conclusions to make them more precise and in line with the results. Furthermore, we specified that the main limitation of our study is given by the number and that further studies are necessary to validate the results (in the article we specified that this is a pilot and exploratory study).

Round 2
Reviewer 2 Report
Comments and Suggestions for Authors
1. Study endpoints should be in methodology part.
2. *statistically significant-It has no meaning. Clarify in legends.
3. The type of stat applied should be mentioned in legends. Tables should be self-explanatory.
4. Table 4- What is indicated by negative and positive sign?
5. Please point out the limitations ethically this will reduce clarity in study.
6. Revise the manuscript thoroughly for both scientific errors and typographical errors.
Thank you
Comments on the Quality of English LanguageDear Editor
The manuscript can be accepted after correcting the remarks.
Thank you
Author Response
- Study endpoints should be in methodology part.
We have moved into the methods. Now it's paragraph 2.1.
- *statistically significant-It has no meaning. Clarify in legends.
We have made the changes. It is now *p value < 0.05.
- The type of stat applied should be mentioned in legends. Tables should be self-explanatory.
We mentioned it into every legends.
- Table 4- What is indicated by negative and positive sign?
In table 4 there are not positive or negative sign. In table 5 and 6 are reported the correlation studied with Pearson test. Pearson’s r is always a number between 0 and 1 and the symbol preceding it indicates the type of relationship between the variables (direct or indirect).
- Please point out the limitations ethically this will reduce clarity in study.
In the conclusions we have explained what we believe are the main limitations of our study. (“Limitations of our study are represented by:
- small sample size;
- impossibility to establish with certainty whether the results observed are linked exclusively to the effect of long-term radiotherapy or to other factors linked to the disease or to the other treatments carried out.
Further studies are required to confirm ours, in order to delve deeper into the pathogenic mechanisms of metabolic syndroms and identify both early diagnosis tools and possible treatments.”)
- Revise the manuscript thoroughly for both scientific errors and typographical errors.
We have carried out a search for errors. We hope we have corrected everything. Thank you